# Extended Applications of the Depth-Sensing Indentation Method

**DOI:** 10.3390/mi11111023

**Published:** 2020-11-23

**Authors:** Dániel Olasz, János Lendvai, Attila Szállás, Gábor Gulyás, Nguyen Q. Chinh

**Affiliations:** 1Department of Materials Physics, Eötvös Loránd University, P.O.B. 32, H-1518 Budapest, Hungary; olasz.dani96@gmail.com (D.O.); lendvai@metal.elte.hu (J.L.); 2SEMILAB Semiconductor Physics Laboratory Co. Ltd., Prielle Kornélia u. 4/a., H-1117 Budapest, Hungary; attila.szallas@semilab.hu (A.S.); gabor.gulyas@semilab.hu (G.G.)

**Keywords:** indentation, phase transformation, plastic instabilities, strain rate sensitivity, indentation creep, grain boundary sliding, dynamic indentation, micropillar-compression

## Abstract

The depth-sensing indentation method has been applied for almost 30 years. In this review, a survey of several extended applications developed during the last three decades is provided. In depth-sensing indentation measurements, the load and penetration depth data are detected as a function of time, in most cases at controlled loading rates. Therefore, beside the determination of hardness and Young’s modulus, different deformation mechanisms and many other dynamic characteristics and phenomena, such as the dynamic elastic modulus, load-induced phase transition, strain rate sensitivity, etc. can be studied. These extended applications of depth-sensing indentation measurements are briefly described and reviewed.

## 1. Introduction

The history of hardness characterization goes back for almost 200 years. Initially, when quantitative characterization of hardness has not yet been defined, qualitative hardness measurement was based on observing which material deforms the other, showing intuitively which material is harder than the other one. A more advanced qualitative hardness measurement, a 10-step scratch hardness scale, still used today in mineralogy, was proposed by Friedrich Mohs in 1822 [1]. Later, more sophisticated, quantitative hardness measuring methods became necessary, which came in the form of indentation tests. The first theory of elastic contact of solids was reported by Hertz in 1881 [2]. In an indentation test a very hard body (indenter) with a well-defined geometry is pushed into the surface of the investigated sample. The hardness is defined as the ratio of the applied force, *F* on the indenter and the projected area, *A* of the contact surface. This surface was measured optically after the removal of the indenter, however, as the demands shifted towards smaller loads, in order to investigate small samples, thin films or to obtain local hardness values, the size of the residual indentations was reduced so that it could no longer be measured with sufficient accuracy by a simple optical method. The development of depth-sensing indentation (DSI) or instrumented hardness measurements, where the indentation load and indentation depth are continuously registered, offered a solution to this problem by methods which used the unloading stage of the indentation curves for determining the size of the residual indentation. Further, it has become possible to determine additional mechanical parameters of materials, such as the Young’s modulus [3]. It should also be emphasized that by setting the loading rate, the average deformation rate of the deformed volume under the indenter can also be controlled, allowing also the study of several characteristics of plastic deformation.

The basic applications of the DSI method are based on the calculations developed by Oliver and Pharr [4,5], which enable the determination of the hardness (*H*) and Young’s modulus (*E*) of the measured material from the loading-unloading indentation curve (Figure 1) without the need of optical measurement of the residual impression. The basic equations of this procedure come from contact mechanics, giving the so-called reduced modulus, *E_r_* as:(1)$$E_{r}=\frac{\sqrt{\pi }}{2}\frac{S}{\sqrt{A}},$$
and the relationship for calculation of the modulus, *E* of the investigated material, as:(2)$$\frac{1}{E_{r}}=\frac{1-\upsilon^{2}}{E}+\frac{1-\upsilon_{i}^{2}}{E_{i}},$$
where $S=\frac{dF}{dh}$ is the contact stiffness measured as the slope of the tangent of the unloading curve at the maximum load, *F_max_*, *A* is the projected area of the elastic contact, *E* and *E_i_* are the Young’s moduli, ν and $\upsilon_{i}$ the Poisson’s ratios of the sample and indenter, respectively. Furthermore, the hardness, *H* of the sample is given by the following equation:(3)$$H=\frac{F_{max}}{A}.$$

During the indentation, the value of the projected area is depending on the contact depth, *h_c_*, where the indenter tip is actually in contact with the surface of the sample. In the Oliver-Pharr evaluating procedure [4], the contact depth can be calculated by the following equation:(4)$$h_{c}=h_{max}-\varepsilon \frac{F_{max}}{S}$$
where $h_{max}$ is the maximum penetration depth measured experimentally, and ε is an indenter-constant (for conical indenter ε=0.72, for Berkovich and Vickers tips ε=0.75, and for flat punch ε=1). For the conventional–conical, Berkovich, Vickers–indenters, the projected area can be given as:(5)$$A=24.5h_{c}^{2}.$$

It should be noted that in fact, Oliver and Pharr [4,5] have improved the suggestions of Doerner and Nix [6], who assumed that the initial part—at *F_max_*—of the unloading curve is linear. Oliver and Pharr, however, have observed that not even the initial part of the unloading curves are linear and they found that the unloading curves can be described rather by a power law function. In this approach, there is no restriction on the unloading data being linear and the contact stiffness is determined only at peak load.

Over the past nearly 30 years, since the publication of the Oliver-Pharr method, DSI has been used to investigate a number of new phenomena and properties in addition to measuring the basic mechanical characteristics. Especially, the introduction of nanoindentation methods has given new impetus to the application of DSI techniques. The purpose of this paper is to present some of the results of (nano)indentation techniques beyond the determination of (nano)hardness and Young’s modulus.

## 2. Extended Applications of the DSI Method

### 2.1. Phase Transformations during Indentation

During a nanoindentation measurement, the high pressure under the tip of the indenter may result in phase transition in the measured sample. The first observation in this direction was reported by Pharr [7] in 1992. Since then, most of the studies regarding this topic were carried out on semiconductors, mainly on silicon [8,9,10,11,12,13,14,15,16], and germanium samples [17,18]. This is due to the wide industrial application of semiconductors, which generates a huge demand on understanding their properties and to the fact that the load inducing phase transformation processes in these materials falls in a range easily attainable in indentation measurements [11].

In general, phase transformations manifest themselves in nanoindentation measurements as pop-out events during the unloading stage [8,10,11,12,13,14,15,16,18,19,20], or as an elbow, which is a change in the slope of the unloading curve [8,10,12,13,14,15,19,20] as shown in Figure 2, where typical load-depth indentation curves indicating phase transition in single-crystal silicon material can be seen.

Since the occurrence of the phenomenon is strongly dependent on the pressure, the most common feature of the phase transformation during indentation is its dependence of the maximum indentation load, *F_max_*. Yan et al. [8], for instance, have carried out indentations using a Berkovich indenter on a single-crystal Si (001) sample with varying *F_max_* between 1 and 200 mN. Based on their measurements the critical maximal load from which pop-out events appeared on the unloading curve was around 30 mN. Below this limit, only elbows were observed. Related transmission electron microscopy (TEM) and selected area diffraction (SAD) investigations [8,10] have detected amorphous silicon (a-Si) in the indented regions where the occurrence of elbows was detected on the *F-h* curves, while in indentations where pop-out events occurred, a mixture of a-Si and crystalline silicon (c-Si) was observed. With higher loads the density of the crystalline phase increased [10].

Domnich and Gogotsi in their review article [19] summarized the high-pressure silicon phases. According to their work, on atmospheric pressure silicon has a cubic diamond structure (Si-I), which under an appropriate load transforms into a β-tin metallic structure (Si-II). By reducing the pressure, the Si-II phase may transform into another metastable rhombohedral structure denoted by Si-XII. With further pressure release a mixture of Si-XII/Si-III structure is created, Si-III being a body-centered cubic structure. These phases can be effectively detected in the residual indents by Raman spectroscopy [9,12,13,15,17,18,19,20]. It has also been shown that the Si-XII and Si-III phases are less dense than the Si-II phase [19], so when the Si-II → Si-XII/Si-III transformation takes place during the unloading, it results in a volume expansion, creating thereby the pop-out [8,10,11,19]. When only an elbow appears on the *F-h* unloading curve, however, the phenomenon is associated with the Si-II → a-Si amorphization process [12,14,19].

High temperature nanoindentation measurements on a Si-I sample can result in the formation of a new phase denoted by Si-IV, which is a hexagonal diamond structure [19]. This kind of transformation can be observed in the temperature range of 350–650 °C [19]. Kiran et al. [15] performed indentation measurements on Si (100) wafers at low (0.2 mN/s) and at high (10 mN/s) unloading rates at elevated temperatures (T = 25, 50, 100, 150, 200 °C). The load-depth curves from their measurements are shown in Figure 3. The pop-outs and elbows appear at different temperatures in the case of the two different unloading rates. The measurements at 200 °C at both low and high unloading rates, however, produce neither pop-outs, nor elbows, at all.

Investigations using Raman microscopy, TEM and electrical resistivity measurements [15] led to the conclusion that with increasing temperature the probability of phase transformation decreases. Electrical resistivity measurements [15], for example, have revealed that the metallic Si-II phase was formed up to 125 °C during loading. Starting from about 150 °C phase transformation as a deformation mechanism loses its importance and deformation induced temperature dependent twinning takes the dominant role.

It should be mentioned that the phenomenon of phase transition during indentation is rather complicated and complex. The appearance of pop-outs and elbows does not only depend on the maximal applied load and on the loading—unloading rates. It may depend also on several other factors, like sample orientation or the indenter-angle etc. [13].

### 2.2. Studying Plastic Instabilities by Indentation

In 1998, investigations of Berces et al. [21,22,23] showed that plastic instabilities can also be studied by analyzing indentation depth-load curves. It has been long established that the strengthening effect of dissolved solute atoms in alloys is brought about by the interaction between solute atoms and dislocations, which hinders the motion of dislocations. In certain cases a dynamic interaction between solute atoms and dislocations leads to the appearance of plastic instabilities, which manifest themselves in the phenomenon of discontinuous yielding during plastic deformation, experimentally observed first by Portevin and Le Chatelier, almost a hundred years ago [24]. Because of its significance in both basic and applied research, these plastic instabilities, often called Portevin-Le Chatelier (PLC) effect, or serrated yielding, or jerky flow [25,26,27,28,29,30,31] have been widely studied in materials science.

The PLC effect has already been observed by several experimental methods, most frequently in tensile tests [32,33,34], but also during compression [35] and torsion [26]. During indentation tests [21,22,23,36,37,38,39,40,41,42,43], the phenomenon of plastic instability appears as steps on the indentation depth–load (*h-F*) curves as shown in Figure 4a in the case of Al-3 wt% Mg and Al-1 wt% Cu alloys. In general, the depth-load relationship is a smoothly changing function, like the curve obtained on pure Al, also shown in Figure 4a. Experimental results [21,22,23,29,30,31,38,41,42] have shown that the occurrence and the development of the instability-steps depend strongly on both the loading rate and the composition of the investigated materials.

Analyzing the step-like indentation depth-load curves, the indentation process can be considered as a series of two elementary steps. In one of these elementary steps-e.g., on the section denoted as AB in the inset of Figure 4a the increase of the load, *F* results in an essentially negligible increase in the indentation depth. In the following elementary step-on the section denoted as BC in the figure a fast and strong increase of the depth *h* takes place at practically constant force, *F*.

According to the depth-load (*h-F*) and depth-time (*h-t*) connections, the corresponding Vickers microhardness, *HV* as a function of the indentation depth, as well as the indentation velocity, $v=\dot{h}$ as a function of the indentation time can be seen in Figure 4b,c, respectively. In the case of smooth depth-load curve, like that obtained on pure Al, the hardness is also smoothly changing as a function of the indentation depth (see Figure 4b). In the case of the step-like depth-load curves of the Al-1Cu and Al-3Mg solid solutions, both the hardness, HV (Figure 4b) and the indentation velocity, *v* (Figure 4c) oscillate quasi-periodically, demonstrating clearly the dynamic strain aging (DSA) [44,45,46], the underlying mechanism of plastic instability in this case. By applying the indentation method, the phenomenon of both the discontinuous deformation—the oscillation in the indentation velocity—and the serrated yielding—the oscillation in hardness—can be observed and studied. Considering the features of both indentation measurements and plastic instabilities, dynamic modeling of the indentation instabilities has been developed [31].

Based on indentation instability measurements, many important results were observed for polycrystalline metallic alloys. It has been shown, for instance, that plastic instabilities occur only if the solute concentration is above a certain critical value. For instance for Al-Mg alloys at room temperature (RT), the critical concentration was found to be 0.86 wt% Mg [36]. Furthermore, in the case of supersaturated AlZnMg(Cu) alloys [29], analyzing the indentation instability, it was concluded that the addition of Cu significantly retards the initial region of the decomposition process—the formation of Guinier Preston (GP) zones—at RT in AlZnMg alloys, the most important family of age-hardenable aluminum alloys.

It should be mentioned that the phenomenon of indentation instabilities as serrated flow has also been observed in several bulk metallic glasses (BMGs) [37,38,39,40,41,42]. However, while in the case of solid solution crystalline alloys the physical basis for the occurrence of indentation instability is the dynamic interaction between mobile dislocation and solute atoms, in the case of BMGs, the serrated flow-pop-in events-is in connection with the formation and propagation of individual shear bands. Despite the essentially different mechanisms, both in crystalline materials and in BMGs the loading rate has a decisive effect on the development of the deformation processes. Figure 5 shows an atomic force microscopy (AFM) image taken on a AlFeGd BMG material indented at different loading rates. Surface steps due to shear bands can clearly be seen around the indents so that the density and the size of the shear bands is depending strongly on the loading rates. The number of shear bands around the indents is lower and the step size is larger at the lower loading rate.

### 2.3. Investigation of Characteristics of Plastic Deformation by Indentation Tests

#### 2.3.1. Determination of Strain Rate Sensitivity by Applying Different Loading Rates

Plastic properties, as the ductility or the mechanism of plastic deformation of materials, are most commonly studied by using tensile tests, often in combination with microstructural methods such as transmission electron microscopy (TEM), X-ray peak profile analysis, atomic force microscopy (AFM), scanning electron microscopy (SEM), etc. Concerning the investigation of the ductility of materials, it is well-established that the strain rate sensitivity (SRS) correlates unambiguously with the ductility [47,48]. Therefore, investigations on the ductility of materials are often focused on investigating the SRS. Several methods were developed for the determination of SRS, including the conventional tensile testing [49,50,51,52], impression creep [53,54,55] and recently in the last decade-depth-sensing indentation testing [56,57,58,59,60]. During an indentation test, by changing the loading rate, the penetration velocity of the indenter, and with it the strain rate can also be changed, leading to a change in the measured hardness, providing a further possibility for the investigation of SRS.

It is well known that in the case of ultrafine-grained (UFG) materials, which can exhibit high ductility, even superplasticity, the mechanism of grain boundary sliding is responsible for both the high ductility and the high SRS [61]. It has been shown that nanoindentation can readily be used to investigate the correlation between these two characteristics of plastic deformation [62]. In this study coarse-grained Al-30Zn sample was ultrafine-grained by using the high-pressure torsion (HPT) technique [63,64], resulting in an UFG structure with an average grain size of about 350 nm. In addition to the grain refinement, HPT processing has also resulted in decomposition of the initial solid solution into AlZn solid solution matrix and high Zn-concentration particles, together with a special structure of the grain boundaries wetted by the segregation of Zn atoms. Because of the grain boundaries wetted by Zn-rich layers, intensive grain boundary sliding can take place in this UFG sample during plastic deformation.

Nanohardness of an UFG Al-30Zn, which exhibits unusually high ductility of about 160% at room temperature [61], was determined by using nanoindentation [62]. In order to study the role of the grain boundaries, measurements were performed under different conditions, on both UFG Al-30Zn and UFG pure Al samples. Series of 400 measurements were performed firstly, at a very low maximum load of 0.5 mN so that the size of the indentation pattern was approximately equal to the average grain size. In this case, the measurements characterize the hardness of individual grains both in the UFG Al and in the UFG Al-30Zn samples. Figure 6 shows the distribution of the nanohardness obtained for two different loading rates on the UFG Al-30Zn sample. It can be seen that the nanohardness distributions are practically the same, indicating that in this case the hardness of the UFG Al-30Zn is not sensitive to the loading rate. This means that when only single grains are deformed, the SRS of UFG Al-30Zn is very low so that the deformation process or the hardness is not sensitive to the strain rate.

Increasing the maximum load of the indentation process to 1 mN, the size of the indentation pattern was ~1–1.5 μm, which is about 3–4 times larger than the average grain size of the UFG Al-30Zn alloy so that the plastic zone under the indentation pattern covers a group of at least ~5–7 grains. The experimental results obtained under this condition, shown in Figure 7a reveal that the distribution of the hardness of the UFG Al-30Zn alloy becomes sensitive to the loading rate. Fitting Gaussian functions to the measured hardness distributions and taking the peak value, similarly to the indentation creep method (see next section) [65] a SRS of about 0.25 is obtained. This proves that when a group of grains is plastically deformed in the UFG Al-30Zn alloy, the role of the grain boundaries–the grain boundary sliding-should be taken into account. Results presented in Figure 7b confirm the low SRS of pure UFG Al as the hardness distribution of this material is independent of the loading rate.

#### 2.3.2. Determination of Strain Rate Sensitivity from Indentation Creep

During the indentation measurements, it is possible to insert an intermediate holding section between the loading and unloading stages. When the load, *F*, at the end of the loading stage reaches its selected maximum value, *F_max_*, the indenter can be held at this maximum load for a required period of time, during which creep deformation may take place in the deformed zone under the indenter. The holding stage is then followed by the unloading, when the load decreases and the indenter moves backwards.

Considering the features of steady-state creep and analyzing the holding stage [58], it has been shown that the depth-time (*h-t*) connection characterizing the creep process during the holding stage can be described by a power-law function, as:(6)$$h=B\cdot {\left(t-t_{c}\right)}^{m/2},$$
where *m* is the strain rate sensitivity (SRS) of the investigated material, *B* and *t_c_* are material constants. Fitting the power-law function in Equation (6) to the experimental data detected during the holding-creep-region, the value of the SRS, *m* can be obtained.

Figure 8 shows some examples demonstrating the validity of Equation (6), and the determination of the *m* parameter for some materials. Find more details in the Ref. [58]. Here we see that the mathematical description of the indentation creep enables a new application of DSI for the determination of SRS of the indented materials.

#### 2.3.3. Demonstrating the Correlation between Strain Rate Sensitivity and Grain Boundary Sliding by Combining Indentation with Atomic Force Microscopy (AFM) Measurements

It was mentioned already in Section 2.3.1, grain boundary sliding is an important mechanism of plastic deformation of UFG materials. Experimental evidences on the grain boundary sliding at RT in UFG metals and alloys have been obtained by combining the indentation and three-dimensional atomic force microscopy (3D-AFM) measurements [61,66,67]. Figure 9 shows an example demonstrating that traces of grain boundary sliding can be observed around indentations on the AFM images [67]. A Vickers indentation pattern can be seen on the surface of an UFG pure Al sample. The AFM image shows very clearly the movement of grains having a size of about 1 μm within the UFG matrix, around the indent.

By combining indentation and AFM, the same phenomenon of intensive grain boundary sliding was also shown in the case of the above mentioned UFG Al-30Zn alloy, which has it’s quite unique plastic behavior due to its superplasticity with a total elongation higher than 150% and a corresponding unusually high SRS (>0.2) at room temperature [61,65]. It was mentioned already that the unique properties of this UFG Al-30Zn alloy at room temperature are due to the sliding of grain boundaries wetted by thin Zn-layers [68,69,70]. In the case of UFG pure Al shown in Figure 9, despite the intensive grain boundary sliding, the value of SRS (≈0.03) and the ductility remains rather low.

Considering the relationship between SRS and the contribution of grain boundary sliding to the total deformation process, calculations have shown that there is no significant difference in the contribution of grain boundary sliding for UFG pure Al and for UFG Al-30Zn. For both materials, this contribution to the total deformation has been measured as 50–60% [61,66,67], despite the significant difference in the SRS. This contradiction has been interpreted by considering the shapes of the pile-ups around the Vickers patterns, recorded by AFM, as shown in Figure 10a, where typical vertical profiles across the Vickers indentation patterns for the UFG pure Al and UFG Al-30Zn can be seen [62,71]. These profiles show that the pile-ups detected around the pattern on the surface of pure Al can be regarded as short-range, appearing sharp only in the close vicinity of the pattern whereas on the surface of the Al-30Zn alloy the pile-ups are rather long-range, spreading over longer distances from the pattern. This difference is caused by the significantly higher mobility of the Zn-wetted grain boundaries in the Al-30Zn alloy [62] so that this alloy becomes superplastic at room temperature. The difference in the flow process of the two materials correlates directly with their SRS, as schematically demonstrated in Figure 10b. In the case of the Al-30Zn alloy, because of its high strain rate sensitivity, pile-ups are not concentrated around the indentation pattern. More details on the correlation between SRS and grain boundary sliding can be found in [62,70,71]. Here we focused on the significance of the combination of indentation and AFM.

#### 2.3.4. Application of Dynamic Indentation for Studying Plastic Behavior

Dynamic indentation can be applied to study viscoelastic systems [72,73,74,75]. In a dynamic indentation measurement, the time-linear $F_{lin}\left(t\right)=vt$ load signal is modulated with a sinusoidal $F_{sin}\left(t\right)=F_{0}\sin\left(\omega t\right)$ component of relatively small amplitude. The resulting penetration depth, *h* is also a harmonic signal on a semi-static depth function, $h_{sin}\left(t\right)=h_{0}\sin\left(\omega t-\varphi \right)$ characterized by a phase shift, φ compared to the load signal [72,75]. Dynamic nanoindentation is also often referred to as continuous stiffness measurement (CSM) [72,76,77,78,79], indicating the continuous calculation of the stiffness, *dF/dh* at the local maximum of the sinusoidal load signal. Combining this local stiffness with the Oliver-Pharr method the hardness and the elastic modulus of the samples can be obtained continuously in the function of the indentation depth [76,77,78].

Recently, Chinh et al. have shown that the phase shift, φ correlates with the SRS of the investigated material [75]. Performing dynamic indentation measurements on coarse- and UFG Al-Zn alloys, Figure 11 shows the main steps of the determination of phase shift, φ and its value obtained on annealed (coarse-grained) and on its UFG counterpart, ultrafine-grained by HPT [75].

Considering also the SRS, *m*, Figure 12 shows the values of *m* as a function of the corresponding φ obtained on the two types of Al-Zn alloys, indicating the correlation between these two quantities. The higher SRS corresponds to a higher phase shift. It should be noted that qualitatively, both quantities, *m* and φ express the rate-dependent properties of the materials. Therefore, this correlation enables a new application of dynamic indentation for studying the rate-dependent deformation-mechanisms of materials.

### 2.4. Indentation for Studying Micro-Plasticity: Compression of Micropillars

As miniaturization progresses, there is an urgent need to understand the deformation processes of small-size samples [68,80,81,82,83,84,85,86,87,88,89,90,91,92]. This is also a typical field, where conventional methods are very hard to apply. Exploiting the advantages of the fine and precise indentation method, in 2003 Uchic et al. [80] introduced a new test methodology for examining plasticity of µm-size samples. For this purpose, focused ion beam (FIB) technology is used to mill so called micropillars of only a few μm diameter, leaving one end of the pillars attached to the bulk sample. During the investigation, the pillar is aligned under a flat-ended indenter in a nanoindentation system and a uniaxial compression test is executed by applying a loading stage.

In their pioneering studies, Uchic et al. [81] have examined the plastic deformation of single-crystal pure Ni pillars of different diameters ranging from 0.5 to 40 μm, as well as the deformation of a bulk sample having dimensions of 2.6×2.6×7.4 mm (Figure 13). Significant differences in the stress-strain curves of the bulk sample and the micropillars (d=5, 10 μm) are clearly visible (see Figure 13A), where much higher yield stress and an intermittent flow in the form of strain bursts were observed in the case of micropillars. It can be seen that in the case of small-size samples, the effect of collective motion of dislocations in individual slip-planes may be very strong, causing catastrophic strain avalanches (concentrated in slip bands) in the samples. In the case of bulk samples, where the flow process is smooth, such avalanches are not observed at all.

It is well-established that this intermittent flow has a stochastic nature [81,82,83] so that both the size of the stress jumps and size of the strain plateaus may differ from sample to sample under the same deformation conditions. Dimiduk et al. have shown that the size of the strain bursts vs. the number of strain burst events follow a power-law function, indicating a scale-free behavior of the avalanches [84]. This statistical nature of the strain avalanches in small-size single crystals has been examined in more details by Ispánovity et al., who have compared the experimental results of compression tests on single crystal Cu micropillars with those of both two- and three-dimensional discrete dislocation dynamic simulations [83].

Kalácska et al. [92] have recently combined micro-pillar compression with three-dimensional high angular resolution electron backscatter diffraction to study the basic deformation mechanism in single crystal Cu of near <100> orientation. This novel combination allows us to follow, for instance, the density evolution and the distribution of the geometrically necessary dislocations as a function of strain, as shown in Figure 14. This new method can certainly give a significant boost to the study of micro-sized samples for understanding the dislocation-controlled micro-scale plasticity.

While the plasticity of micrometer-size single crystal pillars can be characterized by strain avalanches, the properties of fine-grained (UFG) materials are significantly different. Chinh et al. [68] have studied the mechanical properties of UFG Al-30Zn samples having grain-size of 300–400 nm by compressing micro-pillars and found that unlike to their single-crystalline counterparts, no strain avalanches were present in the load-depth curves of the UFG samples (Figure 15). It should be noted that in the case of ultrafine-grained samples, even a sample with a diameter of 3 microns can be considered macroscopic because its diameter is nearly 10 times larger than the average grain-size. For this reason, the property of UFG micropillars in this case reflects the behavior of macroscopic samples.

By scanning electron microscopy investigation [68,85], it was revealed that the stable deformation of the UFG samples is the consequence of intensive grain boundary sliding in this polycrystalline sample. The SEM images of Figure 16, for instance, show significant differences in the surface morphologies of the deformed single-crystal (see Figure 16a,b) and UFG samples (see Figure 16c,d). 

While extreme slip bands can be observed on the surface of the single-crystal pillar, near to the direction of maximum shear stress at 45° relative to the direction of compression, no strain localization and/or extreme slip bands are observed on the surface of UFG pillars. Instead, due to the intensive grain boundary sliding, the motion of individual fine grains leads to deformation having cylindrical symmetry as rings of pushed out grains are formed around the sample, avoiding thereby catastrophic strain avalanches. It was also concluded that because of their high probability of catastrophic failure, single-crystal and/or coarse-grained micrometer-size samples (wires) are not suitable for use in application. The experimental results suggest rather the potential of ultrafine-grained materials in fabrication of micro-devices.

## 3. Summary

More than 100 years after the introduction of quantitative hardness measurement, the invention of depth-sensing indentation measurements and their further development in nanohardness testing have opened up new possibilities in studying the mechanical properties of materials. Depth-sensing measurements routinely allow the determination of the hardness and Young’s modulus of micron-sized samples. In addition, a number of further new research possibilities have been opened up, including e.g., the determination of the hardness and Young’s modulus of even nanoscale components of multiphase systems, due to the high lateral resolution of nanoindenters. Dynamic transformations and mechanical instabilities that occur during indentation can also be studied, and the new area of micromechanics, examining the plasticity of micrometer-size or even smaller bodies, is emerging. In this review, we have presented some examples in which the application of depth-sensing indentation measurements, eventually supplemented by other methods, has led to the study of new mechanical features or phenomena, with the following main points:(1)During a nanoindentation measurement, the pressure under the tip of the indenter may be so high that it may induce structural phase transition in the investigated sample. This phenomenon may manifest itself as pop-out events or elbows appearing on the unloading stage of the indentation curve.(2)The Portevin-Le Chatelier type plastic instabilities, as the phenomenon of discontinuous yielding during plastic deformation, can be studied by using indentation method. During indentation, the phenomenon of plastic instability manifests itself as step-like indentation depth–load curves, indicating discontinuous indentation process. The occurrence and the development of the instability-steps depends strongly on both the loading rate and the composition of the investigated materials.(3)Beside the investigation of plastic instabilities, other dynamic characteristics, such as strain rate sensitivity or viscoelastic behavior can also be studied by using indentation. Furthermore, deformation mechanisms, such as slipping in individual atomic planes, or grain boundary sliding can be observed and investigated by combining indentation with atomic force microscope and/or with scanning electron microscopy.(4)It has also been shown that indentation can be applied for studying micro-plasticity. As today’s devices are getting smaller and smaller, there is an urgent need to understand the deformation processes of small-size samples, where most of the conventional methods are very hard to apply. Typical examples have been shown to demonstrate the advantages of investigating micro-plasticity by compression of micro-pillars using nanoindentation.

## Figures and Tables

**Figure 1 micromachines-11-01023-f001:**
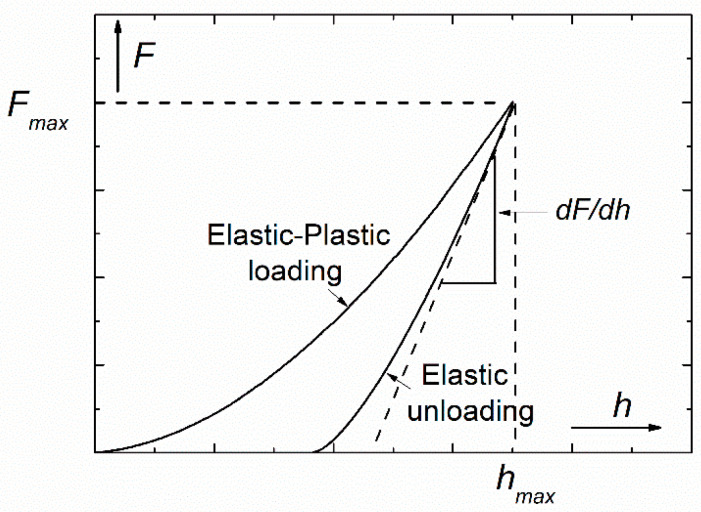
Schematic load-depth (*F-h*) indentation curve with some characteristics necessary for the determination of hardness and elastic modulus.

**Figure 2 micromachines-11-01023-f002:**
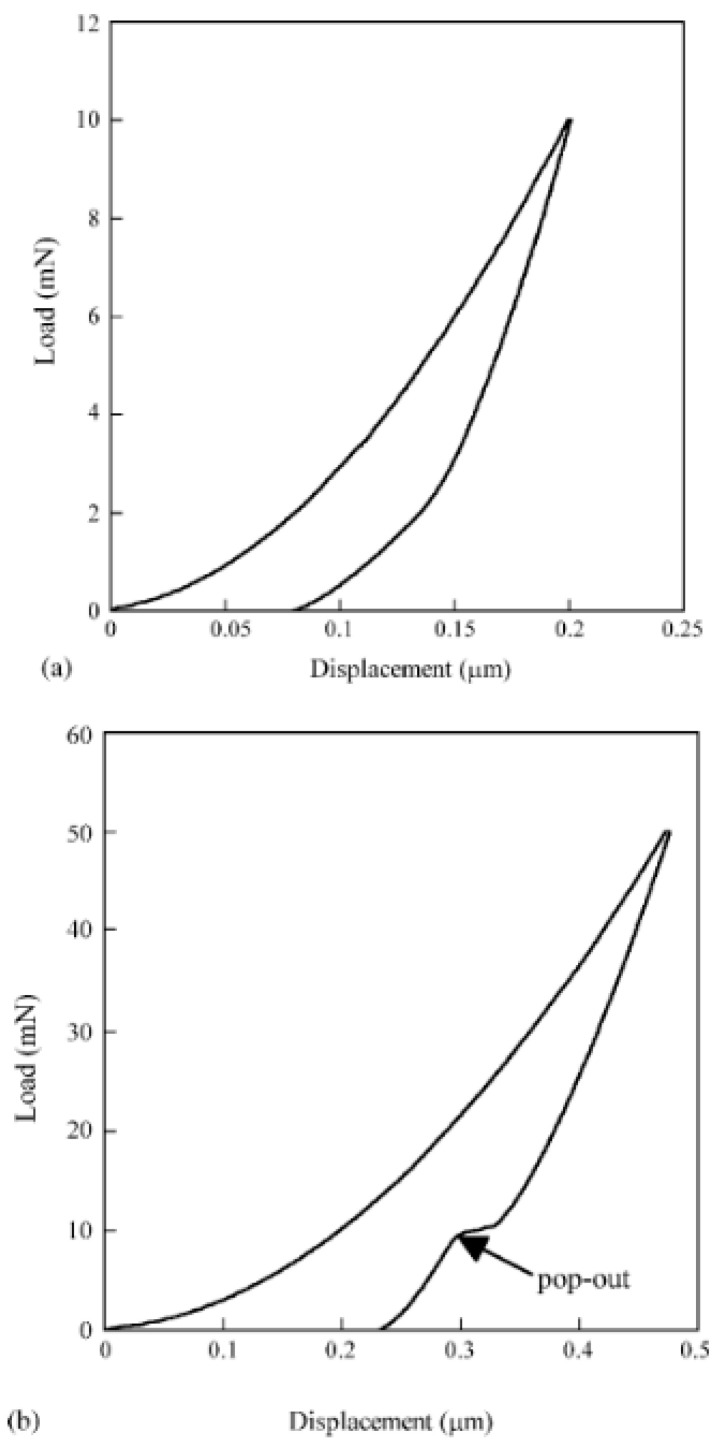
Typical load-depth indentation curve indicating phase transition in a single-crystal silicon wafer sample (**a**) as an elbow at 2 mN on the unloading curve of a measurement up to 10 mN and (**b**) as a pop-out event at about 10 mN in an experiment up to *F_max_* = 50 mN. Reproduced with permission from [8]. Copyright 2006, Elsevier (Amsterdam, The Netherlands).

**Figure 3 micromachines-11-01023-f003:**
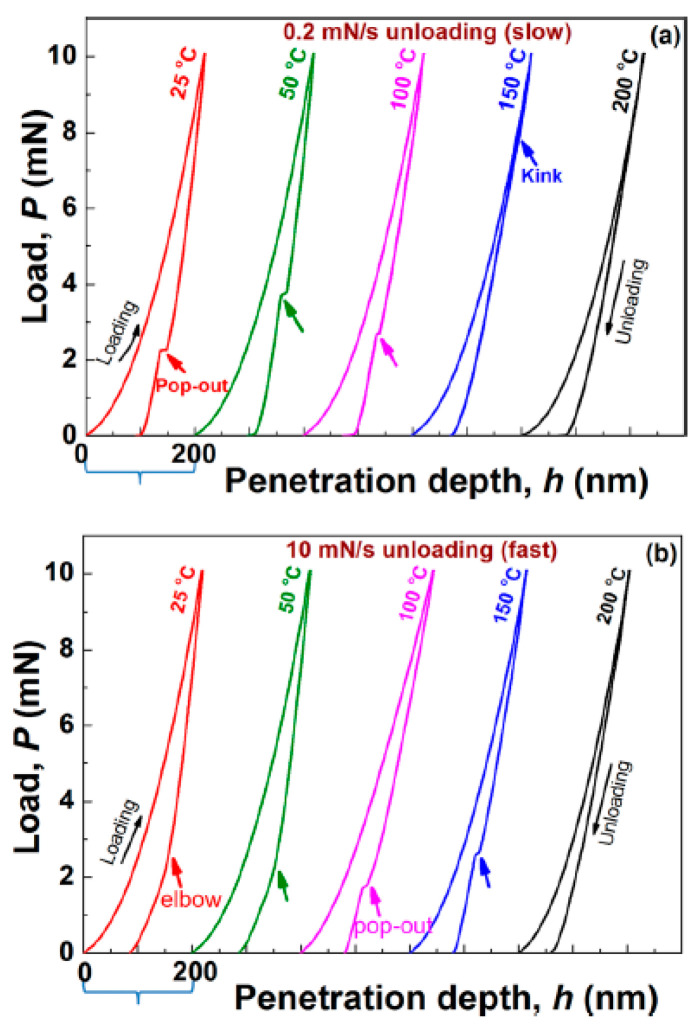
Temperature-dependence of phase-transition during indentation: load-depth curves obtained on diamond cubic silicon (Si-I) at different temperatures for (**a**) 0.2 mN/s and (**b**) 10 mN/s unloading rates. The loading rate of 1 mN/s and holding period of 5 s were the same in all experiments [15]. (In this Figure the load is denoted as P, instead of F.). Reproduced with permission from [15]. Copyright 2015, AIP Publishing LLC.

**Figure 4 micromachines-11-01023-f004:**
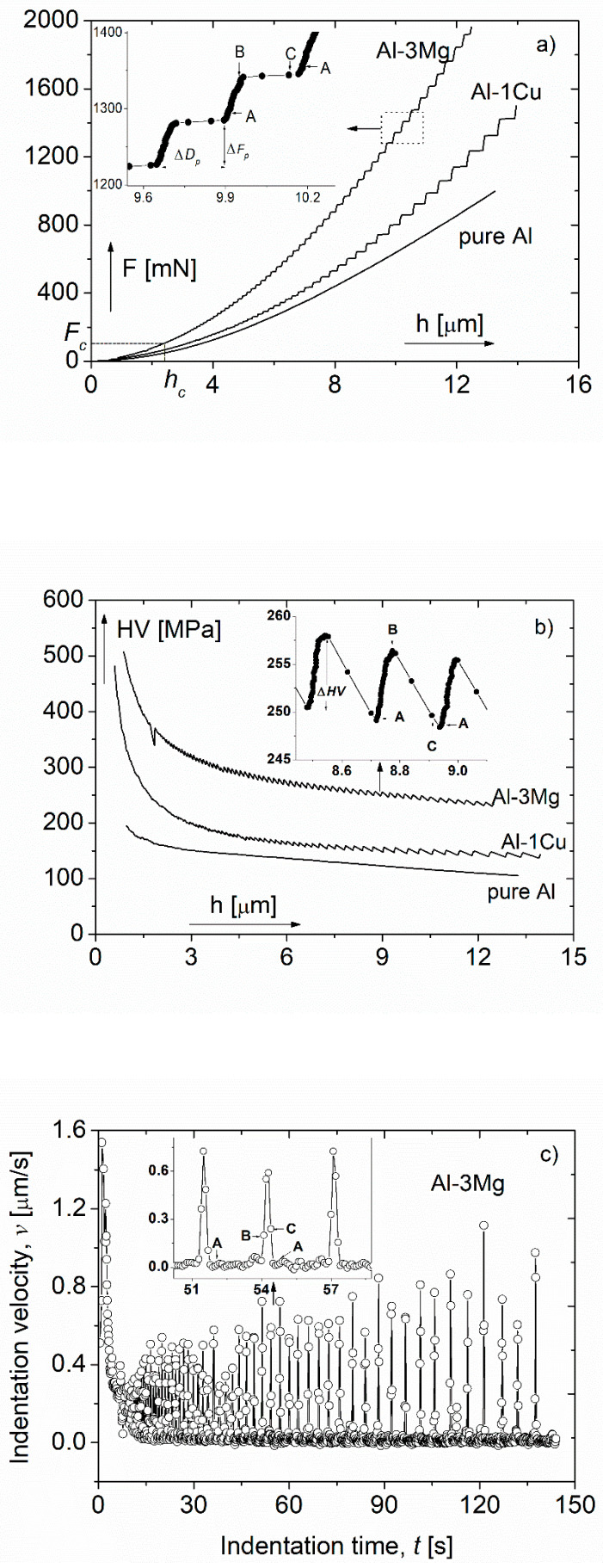
Demonstration of plastic instabilities during indentation: (**a**) Typical indentation depth-load (*h-F*) curves [30], (**b**) changes of the corresponding microharness, HV as a function of the indentation depth, h [30] and (**c**) the changes of the indentation velocity, v as a function of the indentation time, t in the case of the Al-3Mg alloy showing indentation instability [31]. Reproduced with permission from [30,31]. Copyrights 2004, Materials Research Society and 2005, Elsevier.

**Figure 5 micromachines-11-01023-f005:**
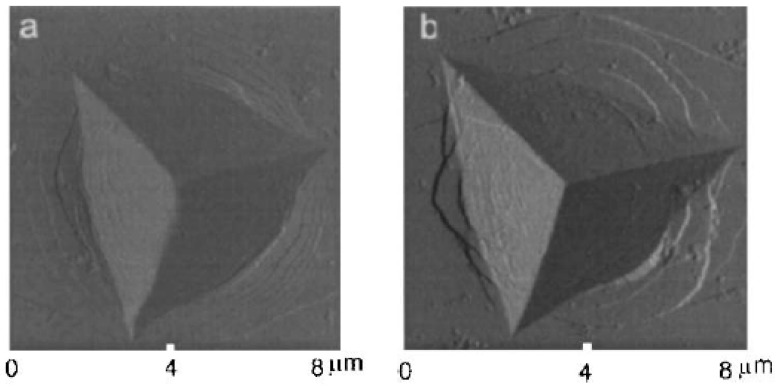
Atomic force microscopy image of Berkovich indents taken at loading rate of (**a**) 100 nm/s and (**b**) 1 nm/s on amorphous AlFeGd ribbon. Reproduced with permission from [30]. Copyrights 2004, Materials Research Society.

**Figure 6 micromachines-11-01023-f006:**
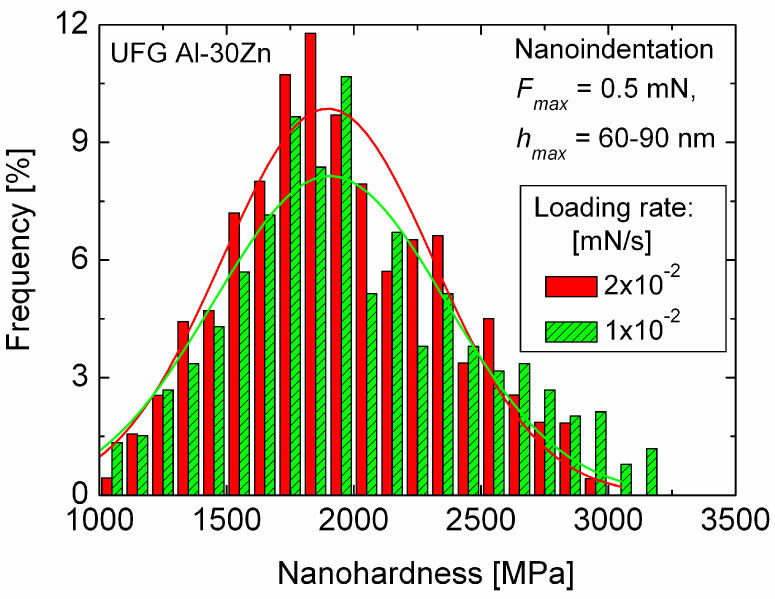
The distribution of hardness obtained at the maximum load of 0.5 mN with two different loading rates on UFG Al-30Zn alloy. The solid lines indicate Gaussian fit of the spectra, showing the very low SRS of the deformation process when practically only single grains are affected by the indentations. Reproduced with permission from [62]. Copyright 2012, Elsevier.

**Figure 7 micromachines-11-01023-f007:**
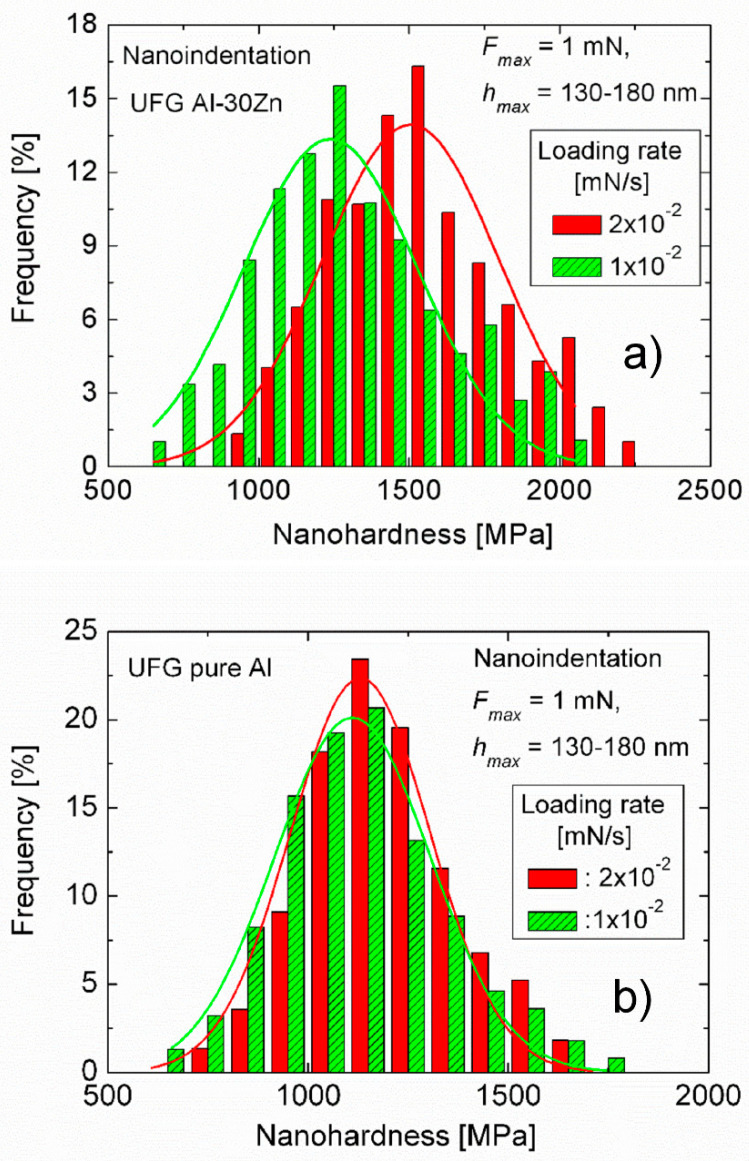
The distribution of nanohardness obtained at the maximum load of 1 mN with two different loading rates on (**a**) UFG Al-30Zn alloy and (**b**) UFG pure Al. The solid lines indicate the Gaussian fit to the distributions, showing the higher SRS of the UFG Al-30Zn sample when a group of several grains is involved in the deformation process. Reproduced with permission from [62]. Copyright 2012, Elsevier.

**Figure 8 micromachines-11-01023-f008:**
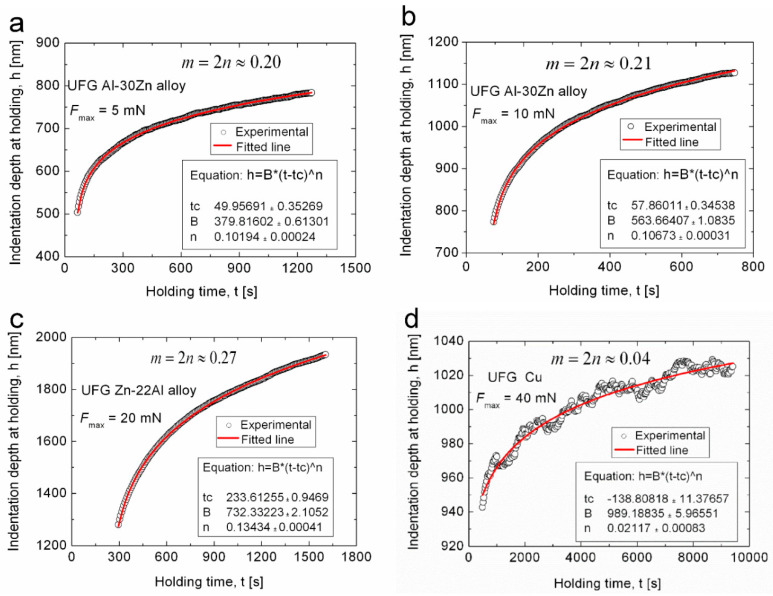
Examples demonstrating the validity of Equation (6) and the determination of the strain rate sensitivity, m. Equation (6) has been fitted to the experimental h-t data obtained at the holding stage for (**a**,**b**) UFG Al-30Zn, (**c**) UFG Zn-22Al and (**d**) UFG Cu. Reproduced with permission from [58]. Copyright 2019, Elsevier.

**Figure 9 micromachines-11-01023-f009:**
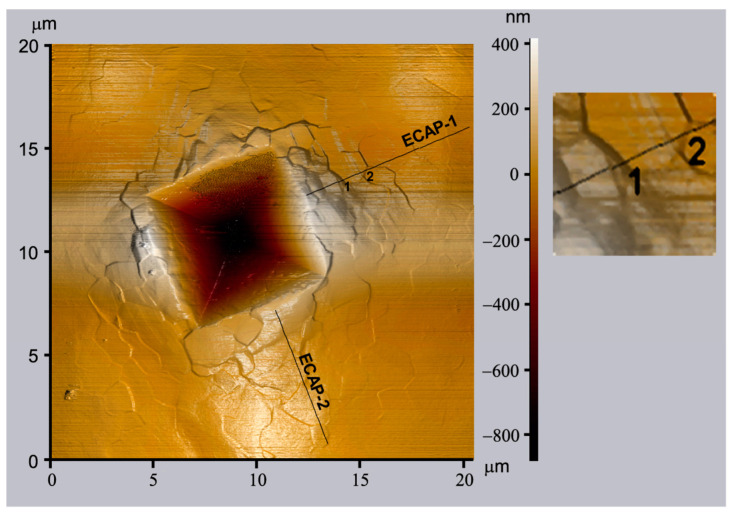
An atomic force microscopy (AFM) micrograph of the surface of an UFG pure Al sample deformed by indentation using a Vickers indenter. Reproduced with permission [67]. Copyright 2006, Elsevier.

**Figure 10 micromachines-11-01023-f010:**
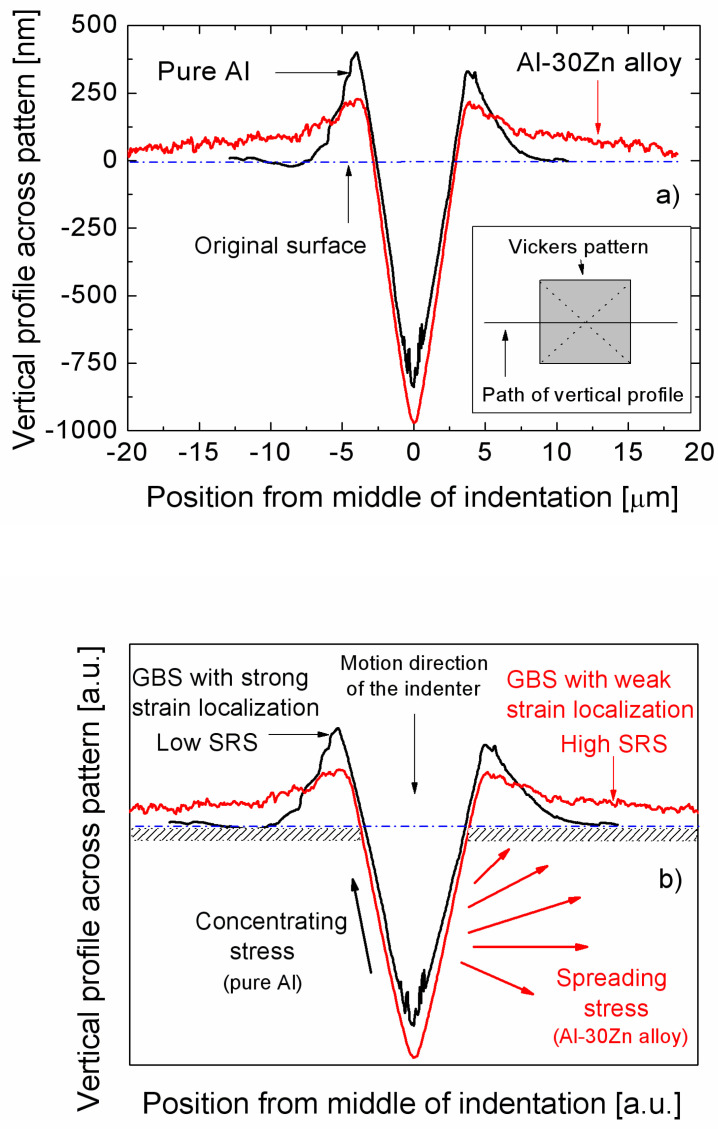
Vertical profiles across the centers of the Vickers indents (**a**) showing the difference in the profiles of UFG pure Al and Al-30Zn and (**b**) schematic picture of the deformation profile under the Vickers indenter and the correlation between grain boundary sliding and SRS. Reproduced with permission from [70]. Copyright 2020, Wiley.

**Figure 11 micromachines-11-01023-f011:**
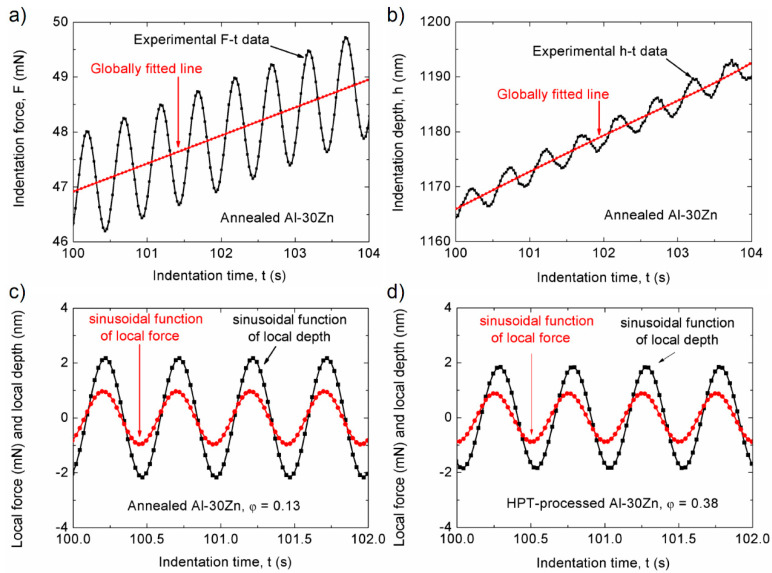
Main steps of the evaluation of the phase shift from the data of dynamic indentation: (**a**) and (**b**) calculation of the oscillatory force and depth signals from the experimental data; (**c**,**d**) the corresponding sinusoidal signals and the phase shift, φ for the annealed coarse-grained and HPT- processed ultrafine-grained Al-30Zn samples. Reproduced with permission from [75]. Copyright 2019, Materials Research Society.

**Figure 12 micromachines-11-01023-f012:**
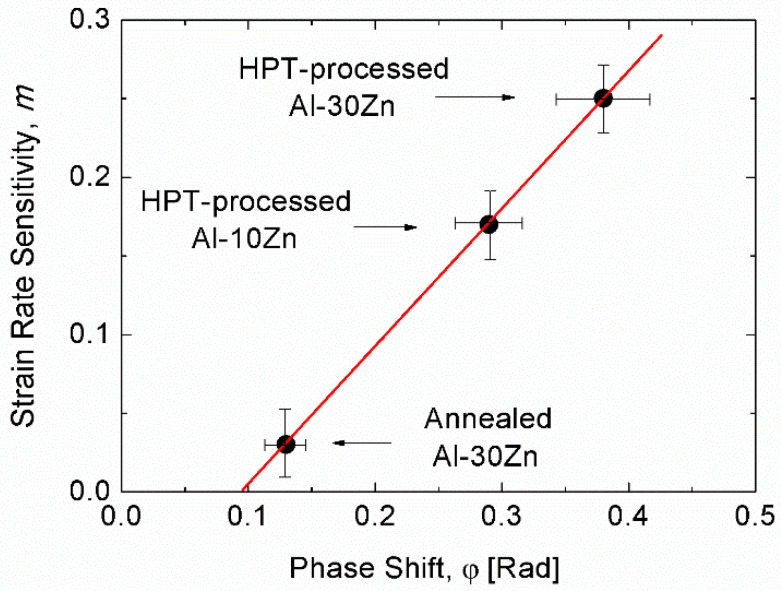
Correlation between the phase shift, φ and the SRS of Al-Zn alloys. Reproduced with permission from [75]. Copyright 2019, Materials Research Society.

**Figure 13 micromachines-11-01023-f013:**
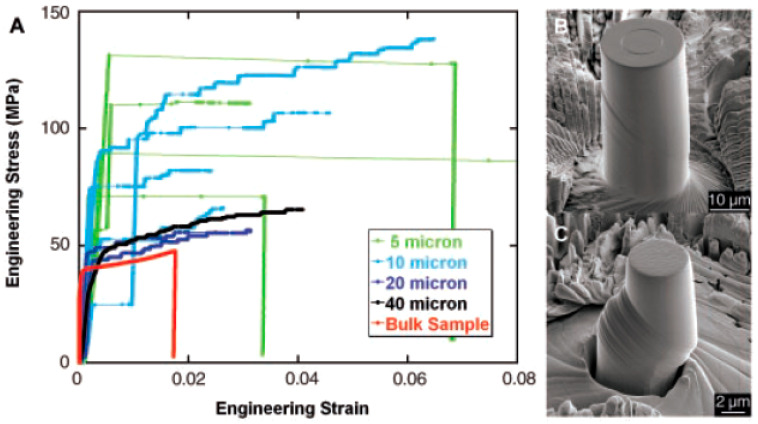
Size-effect in plastic deformation of single-crystal pure Ni micropillars of <134> orientation at room temperature: (**A**) Stress-strain curves for different sample-sizes and (**B**) A scanning electron microscopy (SEM) image of a 20-μm-diameter micropillar deformed to ~4% strain and (**C**) A SEM image of smaller, 5-μm-diameter micropillar, which achieved a rapid strain burst of ~19% strain in less than 0.2 s. Reproduced with permission from [81]. Copyright 2004, AAAS.

**Figure 14 micromachines-11-01023-f014:**
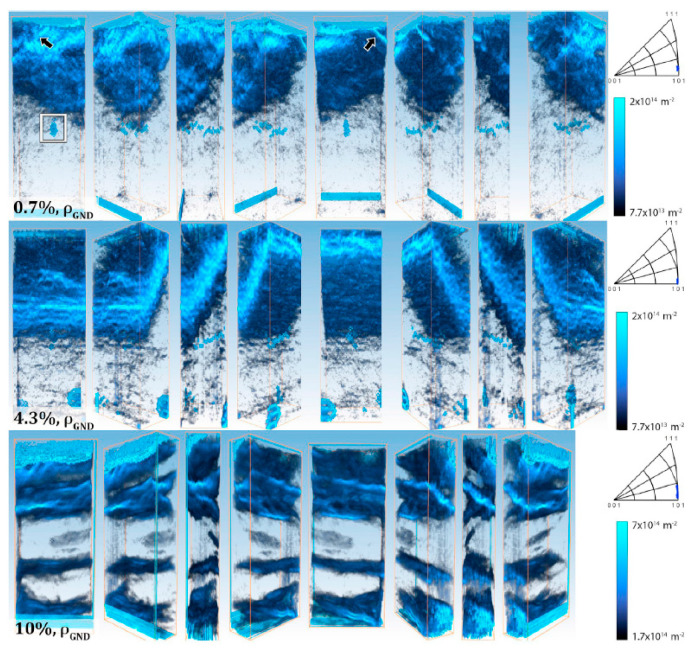
Three-dimensional models of geometrically necessary dislocation density for the three micropillars (the **top**, **middle** and **bottom** rows are referring to 0.7, 4.3 and 10% deformation, respectively), representing an intermediate behavior between bulk and nanoscale plasticity. Well-developed dislocation-cell structure can be observed in the function of deformation. See more details in the Ref. [92]. Reproduced with permission from [92]. Copyright 2020, Elsevier.

**Figure 15 micromachines-11-01023-f015:**
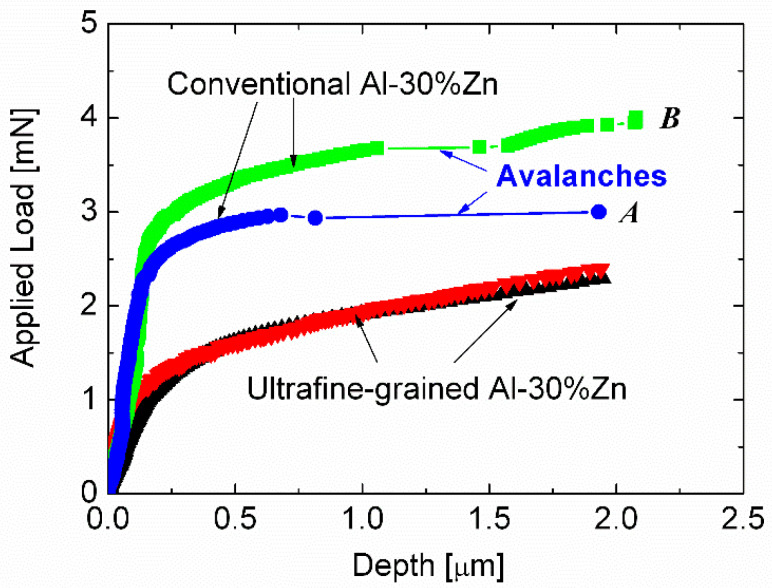
Typical load-depth curves obtained by compressing single-crystal (denoted as conventional) and ultrafine-grained Al-30 wt% Zn micro-pillars, showing typical strain avalanches in the single crystal micropillars, and the avalanche-free, stable deformation of the UFG micropillars. The single crystal pillars **A** and **B** have orientations near <100> and <111>, respectively. Reproduced with permission from [68]. Copyright 2014, Wiley (Hoboken, NJ, USA).

**Figure 16 micromachines-11-01023-f016:**
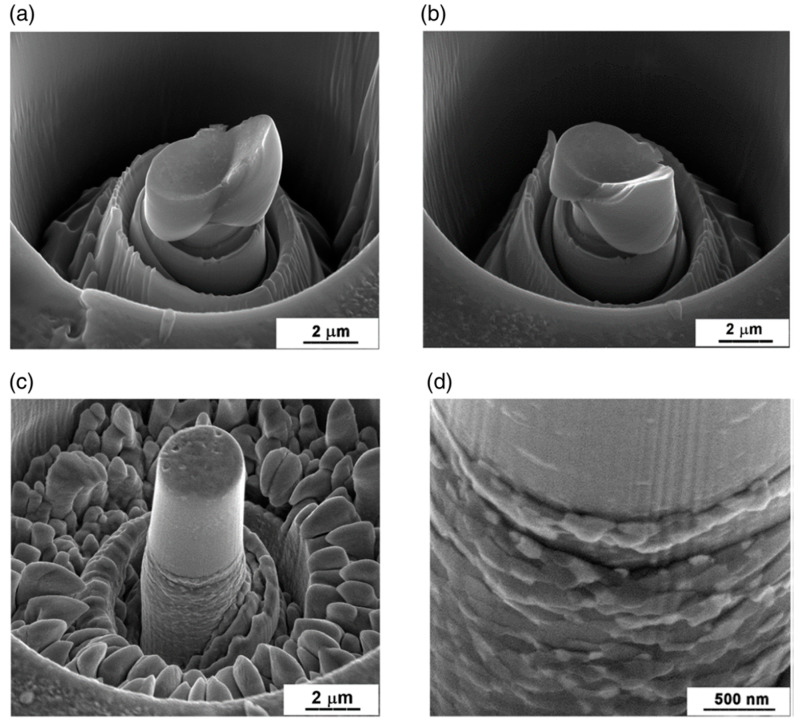
The surface morphologies of (**a**,**b**) single-crystal Al-30Zn sample in two different positions separated by 60° rotation, showing the well-known extreme slip bands corresponding to the avalanches in sample A in Figure 15, and (**c**,**d**) ultrafine-grained Al-30Zn sample at different magnifications, indicating intensive grain boundary sliding without strain localizations and extreme slip bands. Reproduced with permission from [68]. Copyright 2014, Wiley.

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
