# Peer review of "Extended Applications of the Depth-Sensing Indentation Method"

_micromachines, 2020, doi:10.3390/mi11111023_

Round 1
Reviewer 1 Report
The presented paper is a review of extended applications of the indentation method.
I recommend the paper for publication with small changes:
1) The use of word novel to the method has been used for almost 30 years is not necessary. I will suggest changing the title: Extended applications of the depth-sensing indentation method and the first sentence of the abstract: The depth-sensing indentation method has been applied for almost 30 years
2) The first sentence and paragraph of the introduction need to be rewritten and connected with the next paragraph.
3) Some blank spaces need to be removed, dashes and Figure description (Fig or Fig) in the text need to be uniform
4) In Summary I will suggest putting strength on the extended application and remove point 1 as a basic application.
Reviewer 2 Report
The manuscript describes the possible extended applications of depth-sensing indentation method. The presented review would be interesting to materials communities. Thus, the manuscript can be recommended to be published in Micromachines.
The reviewer would like to point out that the review is written at a fairly good level and includes a sufficient number of references.
As a disadvantage of this article, the reviewer can note the lack of information on the application of this method for studying the properties of materials with thin films or coatings, including multilayer ones.
Some comments are also given below.
Lines 139-… - a little confusion arises due to the fact that Figures 2, 3 and 14 are located before the first mention of these figures in the text.
Line 700 - It is necessary to increase the resolution in Figure 13a.
However, these remarks and comments do not reduce the relevance and importance of the article. The article is sufficiently novel and interesting to warrant publication. The list of references reflects the main publications over a fairly long period, on the basis of which the review was written.
